# Advancing PEDV Vaccination: Comparison between Inactivated and Flagellin N-Terminus-Adjuvanted Subunit Vaccines

**DOI:** 10.3390/vaccines12020139

**Published:** 2024-01-29

**Authors:** Asad Murtaza, Nguyen-Thanh Hoa, Do Dieu-Huong, Haroon Afzal, Muhammad Hamza Tariq, Li-Ting Cheng, Yao-Chi Chung

**Affiliations:** 1International Program in Animal Vaccine Technology, International College, National Pingtung University of Science and Technology, Pingtung 91201, Taiwanj10985003@mail.npust.edu.tw (N.-T.H.); dodieuhuong97@gmail.com (D.D.-H.); j11085050@mail.npust.edu.tw (H.A.); 2Department of Biomedical Engineering, Khalifa University of Science and Technology, Abu Dhabi 127788, United Arab Emirates; 100063757@ku.ac.ae; 3Graduate Institute of Animal Vaccine Technology, College of Veterinary Medicine, National Pingtung University of Science and Technology, Pingtung 91201, Taiwan; chenglt@mail.npust.edu.tw

**Keywords:** porcine epidemic diarrhea virus, piglet, flagellin, adjuvant, N-terminus, chimeric protein

## Abstract

Vaccinations can serve as an important preventive measure against the porcine epidemic diarrhea (PED) virus that currently threatens the swine industry. This study focuses on the development of a fusion protein vaccine, FliC_99_-*m*COE, which combines the N-terminus of flagellin (FliC_99_) with a modified core neutralizing epitope (*m*COE) of PEDV. In silico immunoinformatic analysis confirmed the construct’s non-toxic, non-allergenic, and highly antigenic nature. Molecular docking and molecular dynamics (MD) simulations demonstrated FliC_99_-*m*COE’s strong binding to the TLR-5 immunological receptor. Repeated exposure simulations and immunological simulations suggested enhanced cell-mediated immunity. Both FliC_99_-*m*COE and an inactivated PEDV vaccine were produced and tested in mice. The results from cell proliferation, ELISA, and neutralization assays indicated that FliC_99_-*m*COE effectively stimulated cellular immunity and neutralized PEDV. We conclude that the FliC_99_-*m*COE fusion protein may serve as a promising vaccine candidate against PEDV.

## 1. Introduction

In the realm of animal health, there is a pronounced susceptibility to both emerging and re-emerging viral diseases. Notably, the porcine epidemic diarrhea virus (PEDV) has caused significant economic setbacks due to the perpetual emergence of novel variants [1,2]. Classified as an enveloped RNA virus within the Alphacoronavirus genus, PEDV is the etiological agent behind the highly contagious porcine epidemic diarrhea. This disease particularly affects suckling piglets, and its severity is underscored by mortality rates that can reach a staggering 100% [3,4].

The most effective preventative strategy to curb these outbreaks is the development and implementation of potent vaccines. Vaccination of sows prior to parturition facilitates the transfer of maternal antibodies to the fetus, offering protection to newborns [5]. While both live and killed vaccines against PEDV exist, concerns arise due to potential reversion and mutation, posing safety issues. Neutralizing antibody epitopes are crucial for subunit vaccine development. Specifically, for PEDV, neutralizing antibodies serve as an excellent correlate of protection. The spike (S) protein of PEDV is a primary target for these antibodies, especially the COE (collagenase equivalent) region (residues 499–638) which contains most of the conserved neutralizing epitopes [6,7]. Structural studies have also identified the S1D region of spike protein (S) as another target for neutralizing antibodies. The COE domain is prioritized in subunit vaccine development due to its ability to induce neutralizing antibodies in mice [8,9].

Subunit vaccines, owing to their singular molecular composition (usually proteins), necessitate the addition of immunostimulants. Recently, flagellin, a ligand for Toll-like receptor 5 and NOD-like receptor protein 4 as well as the inflammasome receptor NAIP5/6, has been recognized as a promising adjuvant in subunit vaccine research. Our prior work indicates that truncated flagellin can enhance the efficacy of subunit vaccines [10,11,12,13]. The short amino acid sequence *n*FliC (1–99) provides versatility and is amenable to protein expression using the *E. coli* system. Investigations have demonstrated that removal of the hypervariable regions (D2, D3) does not undermine the adjuvanticity of flagellin in vaccine formulations [10].

In prior research, different lengths of flagellin combined with the COE domain (residues 499–638) of PEDV were studied in pigs and mice [8]. Furthermore, inactivated vaccines combined with flagellin were also assessed for efficacy [14]. However, a comparison of immune responses between inactivated vaccines and flagellin-adjuvanted subunit vaccines has not been made. In Taiwan, new PEDV strains like PEDV Pingtung 52 (PEDV-PT) have emerged, and existing vaccines based on PEDV CV777 or DR13 are ineffective against them [15]. An attenuated PEDV-PT vaccine showed promise in 5-week-old piglets, but its production, involving virus isolation and 96 passages, is time-consuming [16].

Drawing from prior research, we sought to assess the adjuvancy of the N-terminus of flagellin (termed FliC_99_, residues 1–99); a chimeric antigen-adjuvant protein was created combining FliC_99_ with a modified COE protein of PEDV named *m*COE (residues 500–799). We conducted in silico analyses followed by vaccine development and subsequent testing in a mice model. FliC_99_ and *m*COE were genetically connected using a glycine–serine (GS) linker. Additionally, we prepared an inactivated PEDV vaccine using the whole virus and compared its immune response to that of FliC_99_-*m*COE.

## 2. Materials and Methods

### 2.1. Structural Investigation

In the computational analysis of the vaccine construct, initial steps involved assessing non-homology with pig proteins using Blastp [17]. The vaccine’s physicochemical properties were then evaluated through the Protparam server, focusing on parameters such as molecular weight, grand average of hydropathicity (GRAVY), stability index, theoretical pI, aliphatic index, and estimated half-life in mammalian cells [18]. Following this, antigenicity and allergenicity were predicted using Vaxijen v-2.0 and AllerTOP, respectively. The final computational step involved using the Psipred 4.0 server to predict the vaccine’s secondary structure, crucial for its folding characteristics and stability [19].

### 2.2. Modeling and Refinement of Tertiary Structure

In this study, the 3D structure of the vaccine construct was proposed using the I-TASSER server (version D-I-TASSER), which models protein structures based on connectivity information [20]. The resulting structure was then assessed using three tools: RAMPAGE for Ramachandran plot construction [21], ERRAT to analyze non-bonded atomic interactions [22], and ProSA-web to verify structural characteristics [23].

### 2.3. Estimation of B-Cell Epitopes

The B-cell epitopes (BCEs), crucial for eliciting humoral immunity in vaccine design, were identified. Conformational and linear BCEs were determined using the Ellipro 1.0.7. tool [24], from IEDB-AR v.2.22, and the ABCPred online tool [25], respectively. For Ellipro, the vaccine’s PDB structure was used as input without altering default parameters. In ABCPred, the vaccine sequence was inputted with a fixed length of 14 residues for epitope prediction.

### 2.4. Molecular Docking Study

In this research, molecular docking analysis was conducted to assess the interaction of the vaccine construct with Toll-like receptor 5 (TLR5), a key target in vaccine design [26]. The vaccine was docked with Sus scrofa domesticus TLR5 using the HADDOCK v.2.2 server, which employs an information-driven flexible docking approach [27]. As the structure of TLR5 was not available on PDB, it was retrieved from Alphafold. The resulting docked complexes were visualized using the PyMOL molecular graphics system, and PDBsum was used to analyze interactions between different residues [28].

### 2.5. Molecular Dynamic Simulations

Molecular dynamics simulations were conducted to validate the stability of the docked complexes. These simulations are essential for understanding protein dynamics and estimating protein stability. The iMOD server, which utilizes internal coordinates normal mode analysis [29], was employed to analyze the overall motion of proteins through internal coordinates. This included assessing internal motion parameters such as covariance, eigenvalues, deformability, and B-factors.

### 2.6. Immunogenicity Evaluation of Construct

The C-ImmSim 10.1 server was used for in silico immune simulations to predict the immunological responses of the vaccine [30]. This server models three critical components of the mammalian immune system: the thymus, lymph nodes, and bone marrow. Default parameters were maintained, except for specific adjustments: the number of injections was set to 2, volume of injection to 1 mL, steps to 100, and random SED to 12,345.

### 2.7. Cell Lines, Virus, and Vectors

Vero cell lines (ATCC) were cultured in minimum essential medium (αMEM, Thermo Fisher Scientific, Waltham, MA, USA) supplemented with 5% fetal bovine serum (FBS, Gibco, New York, NY, USA), streptomycin (100 units/mL), and penicillin (100 mg/mL) in a humidified incubator at 37 °C with 5% CO_2_. PED virus was isolated from locally infected animals, propagated in the Vero cell lines (Appendix A), and purified from the cells that showed a cytopathic effect, and titer was calculated using TCID_50_ assay (50% tissue culture infectious dose) according to the Reed and Muench formula (Reed and Munich 1938). Flagellin of *S. typhimurium* (ATCC 14028) was obtained as a stable plasmid, pET32a-FliC (provided by Prof. Cheng).

### 2.8. Construction of Plasmid

To construct the plasmid FliC_99_, *m*COE, and FliC_99_-*m*COE genes, cloning and ligation into pET32a (Novagen, Darmstadt, Germany) were performed. FliC_99_ (residues 1–99) was cloned using full-length flagellin FliC as a template, with PCR carried out using specific primers (Table 1) before insertion into pET32a. The COE-modified gene of PEDV (residues 500–799) was amplified from the modified S1S2 gene using primers listed in Table 1 and then also inserted into pET32a. For the chimeric FliC_99_-*m*COE construction, a linker was first added to both flagellin and *m*COE through PCR amplification using primers FliC_99_-EcoRI-F, FliC_99_-linker-R, *m*COE-linker-F, and *m*COE-XhoI-R (Table 1). Subsequent chimeric PCR used the FliC_99_ and *m*COE PCR products containing the glycine–serine linker as templates, along with primers from Table 1, before insertion into pET32a. The plasmid constructs were propagated in DH5α (Yeastern Biotech, Taipei, Taiwan), and Sanger sequencing confirmed the FliC_99_-*m*COE-pET32a construct.

### 2.9. Formulation of Inactivated PED Vaccine

Vero cells infected with PEDV (inoculated at a multiplicity of infection of 0.01) were maintained in DMEM containing 5% fetal bovine serum at 37 °C in a 5% CO_2_ atmosphere until a cytopathic effect was apparent. The infected cells were lysed by the freeze–thaw method and centrifuged at 2000× *g* for 10 min. Virions in the supernatant were quantitated and adjusted to 10^7^ median tissue culture infective doses (TCID50)/mL. The inactivated vaccines were prepared through heating at 56 °C for 30 min. The killed virus was stored at −70 °C.

### 2.10. Recombinant Protein Expression and Purification

The recombinant protein FliC_99_-*m*COE was expressed using a method similar to that previously described [10,11]. Briefly, Escherichia coli BL21 (DE3) cells (Yeastern Biotech, Taipei, Taiwan) were transformed with the plasmid construct following the manufacturer’s instructions. The BL21 cells harboring the FliC_99_-*m*COE construct were cultured in Luria–Bertani (LB) broth. Protein expression was induced by adding 1 mM isopropyl-β-D-galactopyranoside (IPTG; Sigma, Darmstadt, Germany) to the culture, followed by incubation at 37 °C for 4 h. Post-induction, the cell culture was harvested, and cells were lysed using native lysis buffer (300 mM KCl, 50 mM KH_2_PO_4_, and 5 mM Imidazole), followed by sonication using an ultrasonic processor (Sonics Vibra Cell, USA). The recombinant protein was then purified using affinity chromatography on Protino^®^ Ni-NTA columns (Bio-scale Mini Profinity IMAC cartridges 1 mL, Bio-Rad, Hercules, CA, USA). The protein was dialyzed in 150 mM PBS to remove urea. Endotoxin levels were assessed using the Pierce™ Chromogenic Endotoxin Quant Kit (Thermo Fisher Scientific, Waltham, MA, USA) and further reduced to 0.25 EU/mL using Pierce™ High-Capacity Endotoxin Removal Spin Columns, 0.5 mL. Following purification, the protein was stored at −20 °C for future use.

### 2.11. Analysis of Recombinant Protein

Recombinant protein expression and quantification were confirmed using 10% sodium dodecyl sulfate–polyacrylamide gel electrophoresis (SDS-PAGE), with bovine serum albumin (BSA) protein standards (KPL, Gaithersburg, MD, USA) for reference. Protein identity was verified by Western blotting using an anti-His-tag monoclonal antibody. The procedure involved loading the protein onto a 10% SDS-PAGE gel, followed by transfer onto polyvinylidene difluoride (PVDF) membranes (Merck, Darmstadt, Germany). The membranes were then incubated with a 6× His tag antibody (GeneTex, Hsinchu, Taiwan) at a 1:5000 dilution as the primary antibody and a rabbit anti-mouse antibody conjugated to horseradish peroxidase (HRP) (GeneTex, Hsinchu City, Taiwan) at a 1:2000 dilution as the secondary antibody. Color development was achieved using Western Lightning PLUS (PerkinElmer, Waltham, MA, USA).

### 2.12. Immunization Schedule and Vaccine Preparation

The FliC_99_-*m*COE recombinant protein was used in a dose-dependent manner at three different concentrations: 2, 5, and 15 μg/mL. The positive control group received an inactivated PEDV vaccine (10^7^ TCID50/mL). Five vaccine formulations were prepared: (1) FliC_99_-*m*COE at 2 μg/mL named 2FC, (2) FliC_99_-*m*COE at 5 μg/mL named 5FC, (3) FliC_99_-*m*COE at 15 μg/mL, named 15FC (4) the inactivated vaccine (IV) at 10^7^ TCID50/mL, and (5) PBS as the control. Each vaccine formulation was mixed with the water-in-oil adjuvant Summit-P101 (SEPPIC, Paris, France) at a 1:1 ratio. Each mouse received a 200 µL dose of the vaccine in all formulations, with injections administered on day 0 and day 14.

Seven-week-old female ICR mice were divided into five groups, with five mice in each. The mice were immunized intramuscularly twice, with a two-week interval between immunizations (Figure 1). Blood samples were collected from a facial vein on days 0, 7, 14, 21, and 28 for immune evaluation. On day 28, all mice (n = 25) were sacrificed to harvest splenocytes for the cell proliferation assay. The experimental protocols involving animals were approved by the Animal Care and Use Committee (NPUST-110-140) at the National Pingtung University of Science and Technology (NPUST). All experiments were conducted in accordance with the ethical rules and laws of NPUST.

### 2.13. ELISA

To determine the titers of total IgG antibodies against PEDV, an indirect enzyme-linked immunosorbent assay (ELISA) was conducted. The serum was isolated from whole blood following centrifugation at 700× *g* for 5 min. A 96-well polystyrene microtiter plate was coated with 100 TCID50/mL of inactivated PEDV coating buffer and incubated overnight at 4 °C. The next day, the plates were washed three times with PBST and blocked with bovine serum albumin (BSA). Then, 100 μL of 2-fold serially diluted serum was added to each well. For detection, a horseradish peroxidase (HRP)-conjugated goat anti-mouse IgG (Sigma, Carlsbad, CA, USA) was used at a 1:5000 dilution. Tetramethylbenzidine (TMB, Sigma) served as the chromogen for color development. The reaction was halted using 2 M H_2_SO_4_, and absorbance was measured at 450 nm using an MultiskanTM FC microplate photometer (Thermo Fisher Scientific, Vantaa, Finland).

### 2.14. Lymphocyte Proliferation Assay

To assess T-cell proliferation activity, a CCK-8 assay was performed. Mice were sacrificed on day 28, and spleens were extracted and placed in RPMI 1640 medium (Gibco Invitrogen, Carlsbad, CA, USA) containing antibiotics. The spleens were gently mashed with a plunger, and the fluid containing splenocytes was aspirated and filtered to remove larger tissues. The splenocyte-rich fluid was then centrifuged at 1500 rpm, 4 °C for 10 min, and the supernatant was discarded. ACK buffer (0.07 M NH_4_Cl, 0.07 M Na_2_CO_3_, and 0.01 M EDTA at pH 7.4) was added to lyse red blood cells. After two washes in RPMI 1640, cells were resuspended in RPMI 1640 supplemented with 10% FBS (fetal bovine serum) and 1% antibiotic (penicillin and streptomycin) (Gibco Invitrogen, Carlsbad, CA, USA) at a concentration of 1 × 10^6^ cells/mL.

For the assay, splenocytes (1 × 10^5^ cells/well) were seeded in 96-well plates and incubated with 5 μg/mL of purified recombinant *m*COE for 24 h at 37 °C and 5% CO_2_. Concanavalin A (Thermo Fisher Scientific, Ward Hill, MA, USA) at 5 µg/mL served as a positive control, while RPMI 1640 was the negative control. The next day, CCK-8 solution was added and incubated for 2 h. Cell viability was measured using the Cell Counting Kit-8 (Dojindo, Rockville, MD, USA), with absorbance read at 450 nm using the MultiskanTM FC microplate photometer (Thermo Fisher Scientific, Vantaa, Finland). The stimulation index (SI) was calculated as follows: (OD of the treatment − OD of background)/(OD of the negative control − OD of background).

### 2.15. Virus Neutralization Test (VNT)

Virus-neutralizing antibody titers were determined using PEDV and Vero cell lines. Initially, Vero cells were cultured at a density of 5 × 10^5^/mL in a 96-well cell culture plate using minimum essential medium (MEM) (Gibco, Thermo Fisher) supplemented with 5% FBS and 1% antibiotic. The cells were incubated for one hour at 37 °C. Serum samples, heat-inactivated and diluted in a Log_2_ series, were mixed with 1000 median TCID50/mL of PEDV. This virus–serum mixture was then added to the Vero cell cultures (in triplicate for each serum dilution) in 96-well plates and incubated at 37 °C in a 5% CO_2_ environment. After five to seven days, the cell monolayers were inspected for a cytopathic effect (CPE). The complete absence of a CPE was recorded as successful virus neutralization.

### 2.16. Statistical Analysis

For statistical analysis of the results, Graph Pad Prism (version 9) was utilized. A two-way ANOVA was applied to compare antibody responses and virus neutralization test data. The Mann–Whitney test was used for lymphocyte proliferation comparison studies. A significance level of 0.05 was set for all experiments, and data are presented as mean ± standard Error.

## 3. Results

### 3.1. Physiochemical and Immunogenic Profiling

ProtParam analysis revealed that the vaccine’s theoretical pI is 4.92, with a molecular weight of 43,131.08 Daltons. The aliphatic index was calculated as 82.05, and the GRAVY as 0.024. The predicted half-life of the vaccine was estimated at 30 h in mammalian reticulocytes in vitro, over 10 h in *Escherichia coli*, and over 20 h in yeast cells in vivo. Homology assessment with the animal’s proteome showed no significant similarity, indicating minimized risk of autoimmunity. Additionally, Vaxijen and Allertop online servers confirmed the vaccine’s high antigenicity and low allergenicity. Collectively, these properties suggest that the construct has substantial potential as a viable vaccine candidate against PEDV.

### 3.2. Structural Assessment

The secondary structure of the vaccine construct was predicted using the I-Tasser online tool. It was found that 184 amino acids (45.6% of the construct) formed random coils, constituting the most prevalent structural feature. Beta strands, composed of 131 residues (32.4%), were the second most common structure. The least common was the alpha helix, with only 89 residues (22%) forming this structure (Appendix A).

### 3.3. Modeling of Tertiary Structure

The 3D structure of the vaccine was predicted using the I-Tasser server and visualized with Discovery Studio (Figure 2a). Ramachandran plot analysis (Figure 2b) indicated that 345 out of 404 amino acids (85.3%) were in favored or additionally allowed regions, while only 11 amino acids (2.7%) were in disallowed regions. Further validation revealed no weak rotamers in the predicted structure, as evidenced by a Prosa-web Z-score of −6.12 (Figure 2c) and an ERRAT quality factor of 78.41. These results collectively confirm the high quality of the modeled structure.

### 3.4. Selection of B-Cell Epitopes

The presence of B-cell epitopes (BCEs) on the vaccine structure is crucial for inducing humoral immunity through B-lymphocyte-mediated antibody generation. In our vaccine construct, BCEs were identified using the Ellipro and ABCPred servers. This analysis revealed that the vaccine contains 37 linear/continuous BCEs (LBL) and 6 conformational/discontinuous BCEs (CBL), as detailed in Appendix A, respectively.

### 3.5. Molecular Docking

Molecular docking studies to examine the interactions between TLR5 and our vaccine construct were conducted using the HADDOCK server. The results, presented in Table 2, show a strong binding score of 480.6 ± 22.4 kcal/mol and a root mean square deviation (RMSD) of 12.9 ± 0.4. Additionally, PDBsum analysis revealed the formation of 16 hydrogen bonds between the ligand (vaccine construct) and the receptor (TLR5), with bond lengths within 3.31 Å, as illustrated in Figure 3 and detailed in Table 3.

### 3.6. Molecular Dynamic Simulation

Normal mode analysis (NMA) using the iMODS server was utilized to evaluate the stability of the docked complex. This analysis considers the internal coordinates of the complex to assess protein mobility and stabilization. The deformability was determined by analyzing distortions in amino acid residues, depicted as chain hinges (Figure 4a). Eigenvalues were calculated as the reciprocal of the variance of each normal mode, with the overall eigenvalue for the complex being 1.500138 × 10^−6^ (Figure 4c). Additionally, the B-factor, proportional to RMS in NMA, was assessed (Figure 4b). An elastic map visualizing atoms connected by springs was generated (Figure 4d), where each spring is represented by a gray dot, with denser dots indicating stiffer regions. Finally, a covariance matrix was created to identify correlations between amino acid residue pairs, using white, blue, and red to represent irrelevant, disassociated, and correlated motions, respectively (Figure 4e).

### 3.7. Immune Simulation

Once the body is infected with a pathogen, primary and secondary immune responses will be mounted. Activation of both of these responses is important for a vaccine to elicit effective immunity. In this study, immune simulations were performed using in silico methods to predict immune responses of the vaccine. As shown in Figure 5, the vaccine is predicted to result in the elevated production of IgM, IgG, IgG1, and IgG2. The detailed analysis of the immune response is presented in the Appendix A.

### 3.8. Generation of Subunit Vaccine and Characterization

The FliC_99_-*m*COE fusion protein (Figure 6C) was created by combining FliC_99_ (Figure 6B) with the *m*COE of PEDV (Figure 6A) using a glycine–serine linker. This protein was expressed in *E. coli* and its presence was confirmed through SDS-PAGE (Figure 7A) and Western blot analyses (Figure 7B), which showed a molecular weight of 63 kDa. Additionally, the recombinant proteins included a 20 kDa Trx-His-S-enterokinase tag from the pET32a vector, contributing to the overall size. The protein was purified multiple times (detailed in the Appendix A respectively), resulting in a final concentration of 106 µg/mL (Figure 7C).

### 3.9. PEDV-Specific Antibody Titers Increased by Flagellin-Adjuvanted and Inactivated PED Vaccines

Antibody responses in immunized mice were assessed via indirect ELISA, employing *inactivated PEDV* as the coating antigen. Overall, the serum antibody titers in the group immunized with the inactivated vaccine were significantly higher at all time points compared to the flagellin-adjuvanted PED vaccine group. Notably, the antibody response in the flagellin-adjuvanted PED vaccine groups increased in a dose-dependent manner, with the 2FC group showing the lowest and the 15FC group the highest response (Figure 8). However, no significant differences were observed among the flagellin-adjuvanted PED vaccine groups (2FC, 5FC, and 15FC).

### 3.10. Both Flagellin-Adjuvanted and PED Vaccine Candidates Proliferate the Lymphocytes In Vitro

Lymphocyte proliferation was measured using splenocytes extracted from all immunized mice groups. The splenocytes were stimulated using *m*COE, and the proliferation activity was quantified. In all three flagellin-adjuvanted PED vaccine combinations (2FC, 5FC, and 15FC), proliferation activity increased significantly compared to the negative control (Figure 9A). Notably, the 5FC group exhibited the highest proliferation activity within the flagellin-adjuvanted PED vaccine group. Among all groups, the inactivated vaccine group showed the highest proliferation activity. Additionally, the proliferation activity in the positive control was significantly higher compared to the negative control (Figure 9B). These results indicate a substantial increase in lymphocyte numbers due to the addition of flagellin.

### 3.11. Flagellin-Adjuvanted and Inactivated PED Vaccine Groups Elicited Neutralizing Antibodies in Mice

On day 28 post-first immunization, serum samples were analyzed for serum neutralizing antibody titers. Both flagellin-adjuvanted and inactivated PED virus vaccine groups demonstrated the ability to neutralize the virus. The inactivated vaccine group exhibited a significantly higher level of neutralizing antibodies, achieving an antibody titer of 4.5 Log_2_. In contrast, all flagellin-adjuvanted groups (2FC, 5FC, and 15FC) reached a neutralizing antibody titer of 3.5 Log_2_, with no significant differences observed among these groups. Conversely, the PBS group failed to neutralize the virus, as shown in Figure 10.

## 4. Discussion

The current study demonstrates that conserve neutralizing domains (COE, SID) of PEDV show potential as a vaccine antigen when linked to flagellin (1–99), showing comparable results with the inactivated PEDV vaccine.

Computational methodologies in vaccine development offer numerous advantages, including rapid antigen identification from extensive databases, epitope prediction, and the prediction of a vaccine’s efficacy, stability, and immunogenicity, as well as aiding in formulation optimization [31]. These approaches facilitate the efficient, cost-effective identification of promising vaccine candidates, advancing the development of safe and effective vaccines against diverse pathogens [32]. Recently, such computational approaches have been used to propose drugs and vaccines against a wide range of viruses such as SARS-CoV-2 [33], dengue virus [34], Hendra virus [35], Clostridioides difficile [36], and others. In this study, similar computational strategies were employed to predict the antigenicity, allergenicity, and toxicity of our vaccine candidate. Additionally, the potential to provoke immune responses was estimated through molecular docking, molecular dynamics (MD) simulations, and immune simulations.

When evaluating the physicochemical properties of our vaccine candidate, the molecular weight was determined to be 43.14 kDa. This is within an acceptable range, as proteins lighter than 100 kDa are typically easier to express and purify [37]. The theoretical isoelectric point (pI) was 4.92, indicating the vaccine’s acidic nature. Notably, the vaccine demonstrated extended half-lives of over 20 h in *E. coli* and over 10 h in yeast cells, suggesting prolonged exposure to the immune system. The vaccine’s stability index was less than 40, indicative of high stability under standard conditions [18]. Additionally, a high aliphatic index of 82.05 points to the vaccine’s stability across a broad temperature spectrum [38]. Therefore, the vaccine’s combination of a favorable instability index (34.72) and a high aliphatic index confirms its stability both under standard conditions and across various temperatures.

To effectively combat infections, both cellular and humoral immunity play vital roles [30], with humoral immunity primarily mediated by antibodies produced by B-cells [39]. In vaccine design, discontinuous B-cell epitopes (BCEs) are crucial for triggering antigen–antibody reactions [40]. Given that our vaccine candidate exhibited a high number of such discontinuous BCEs, it is poised to potentially induce a robust humoral immune response.

TLR5 plays a critical role in vaccine development, particularly in the context of computational approaches, due to its ability to recognize flagellin and initiate immune responses. Computational methods enable the identification of TLR5 agonists, the development of effective adjuvants, and the integration of TLR5-targeting components into rational vaccine design. As such, vaccines demonstrating binding affinity toward TLR5 are likely to elicit potent immune responses against viral infections [41]. To determine the potential of our vaccine candidate to bind with TLR5, we conducted molecular docking and molecular dynamics (MD) simulation studies. These methodologies have been previously utilized to explore interactions between vaccines and ligands in various studies [42,43]. The molecular docking results indicated that the complex formed between TLR-5 and our vaccine construct is potentially effective in eliciting both cellular and humoral immune responses, as evidenced by a docking score of 480.6 ± 22.4 and an RMSD value of 12.9 ± 0.4. Additionally, molecular dynamics (MD) simulations reinforced the stability of this complex. Complementing these findings, in silico immune simulations suggested that the vaccine is capable of efficiently triggering both adaptive and innate immune responses, with a rapid clearance observed upon subsequent exposures.

Upon administration of the PEDV subunit vaccine, we observed a dose-dependent increase in antibody response, accompanied by a rise in lymphocyte proliferation. Notably, the inactivated vaccine consistently elicited stronger humoral and cellular immune responses throughout the various immunological evaluation periods [14], despite the lack of statistically significant differences between FliC_99_-*m*COE immunization groups. Interestingly, both the subunit and inactivated PEDV vaccines reached peak neutralizing antibody titers concurrently [8]. Our study did not include a challenge test in mice due to their resistance to PEDV. Therefore, the ideal evaluation of vaccine efficacy should involve immunizing pregnant sows and challenging their suckling piglets with PEDV [8]. This finding indicates that even a low dose of the subunit vaccine can achieve a protective response comparable to higher doses, which is particularly advantageous given the inclusion of flagellin as an adjuvant [44]. This suggests an efficiency of the vaccine formulation that could have significant implications for its practical application and cost-effectiveness.

The inactivated PEDV vaccine, encompassing all key pathogenic factors, effectively elicits an immune response. However, this approach faces challenges such as unintended immune activation and the need for multiple boosters to achieve a protective immune response. In contrast, our FliC_99_-*m*COE construct, comprising select immunogenic epitopes of the S-protein linked to a short flagellin adjuvant [11], demonstrates a marked increase in neutralizing antibodies and other immune factors, showing comparable efficacy to the inactivated PEDV vaccine. Crucially, the FliC_99_-*m*COE presents itself as a viable alternative to conventional inactivated PEDV vaccines, retaining the valued safety profile typical of subunit vaccines and potentially reducing the requirement for multiple immunizations.

The flagellin N-terminus, owing to its small size and role as a “danger signal” through TLR5, significantly enhances the efficacy of the PEDV subunit vaccine. Adjuvants like flagellin are known to augment antigen-specific immune responses and facilitate efficient antigen delivery to dendritic cells. Li et al. pioneered the development of the first COE–flagellin chimeric vaccine, recognizing the synergistic potential of the S-protein and flagellin. In their study, the flagellin–COE group exhibited elevated levels of cytokines and neutralizing antibody titers [8]. In subsequent experiments, they used a COE–flagellin chimeric subunit vaccine, alongside flagellin and COE alone, for immunizing pigs. Their observations revealed enhanced neutralizing antibody and cytokine responses in the COE–flagellin group, coupled with superior protection against highly pathogenic PEDV challenge compared to the other groups [45]. Building on this concept, our current work involves coupling a truncated flagellin to *m*COE, yielding an immune response with neutralizing antibodies comparable to that produced by an inactivated vaccine. Previous research has validated the efficacy of N-terminal flagellin (residues 1–99) as an adjuvant in vaccines against infectious bursal disease virus (IBDV), PlpE, duck hepatitis virus, and Actinobacillus pleuropneumoniae. These studies collectively affirm the adjuvant potential of truncated flagellin [10,11,12,13].

Future research should emphasize the oral administration of the PEDV subunit vaccine, which holds considerable promise in controlling PEDV. This virus primarily targets mucosal surfaces in the intestine, affecting M-cell numbers and intestinal morphology [46]. Effective mucosal delivery of the PEDV subunit vaccine could potentially prevent virus entry and propagation [47]. Flagellin itself is a great activator of mucosal immunity but requires surface encapsulation to survive in microbiota. However, encapsulation is required to achieve potential vaccine advantages against PEDV. A future study design targeting an oral route is warranted to test flagellin-*m*COE’s potential.

## 5. Conclusions

In conclusion, this study introduces a promising PEDV subunit vaccine combining a modified epitope (*m*COE) with truncated flagellin, eliciting strong immune response and safety in mice. Computational analyses, including molecular docking and MD simulations, indicate strong TLR-5 affinity, enhancing immunogenicity. In vivo studies in mice reveal an efficacy comparable to inactivated vaccines. These findings advocate the vaccine’s potential, highlighting the synergy of computational and empirical approaches in vaccine development against PEDV.

## Figures and Tables

**Figure 1 vaccines-12-00139-f001:**
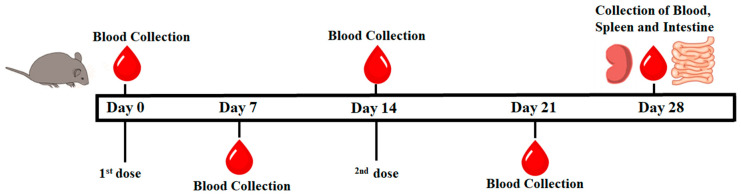
Design of the mice immunization process.

**Figure 2 vaccines-12-00139-f002:**
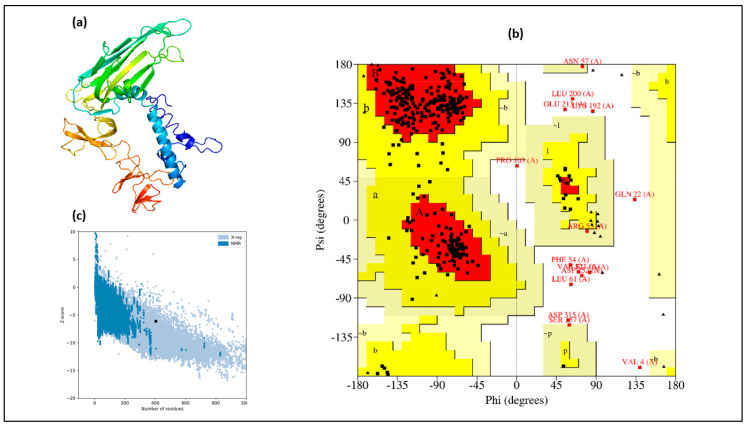
(**a**) I-Tasser modeled tertiary structure of vaccine. (**b**) Ramachandran plot of predicted structure, (**c**) Prosa-web-made Z-score graph of vaccine structure.

**Figure 3 vaccines-12-00139-f003:**
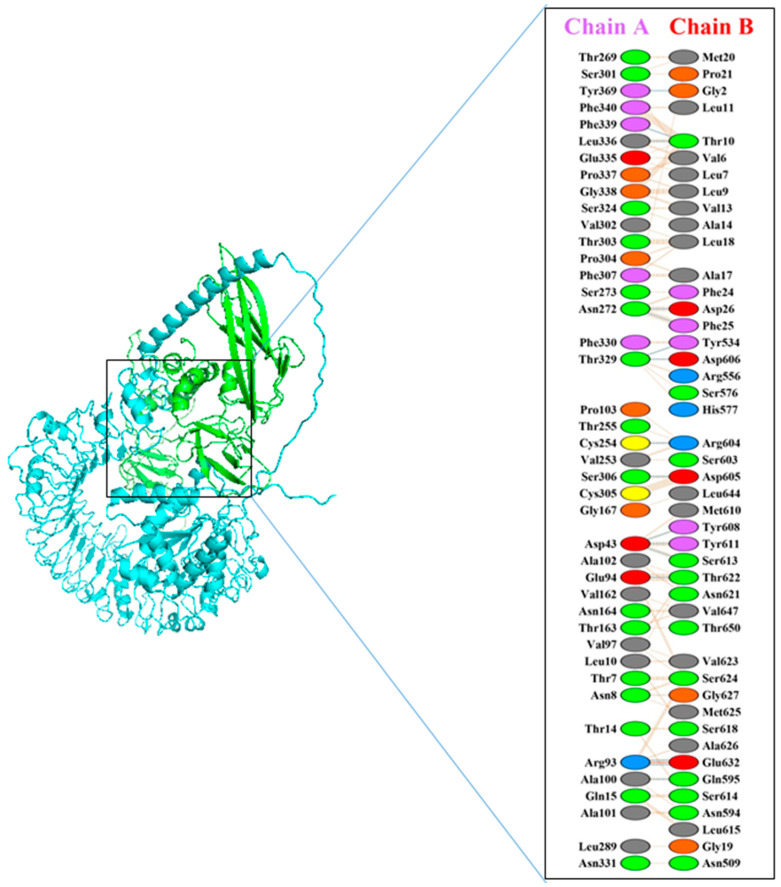
Docked complex of TLR5–vaccine construct in the form of cartoon representation, with TLR5 in blue and vaccine in green. Furthermore, Interacting residues of complex are shown, where blue lines depict H- bond, red lines demonstrate salt bridges, and orange lines represent other interactions. The color scheme of amino acids is as follows: green = neutral (side chains with neither acidic nor basic properties), pink = aromatic, blue = positively charged, red = negatively charged, orange = glycine and proline, and gray = amino acids with aliphatic side chains.

**Figure 4 vaccines-12-00139-f004:**
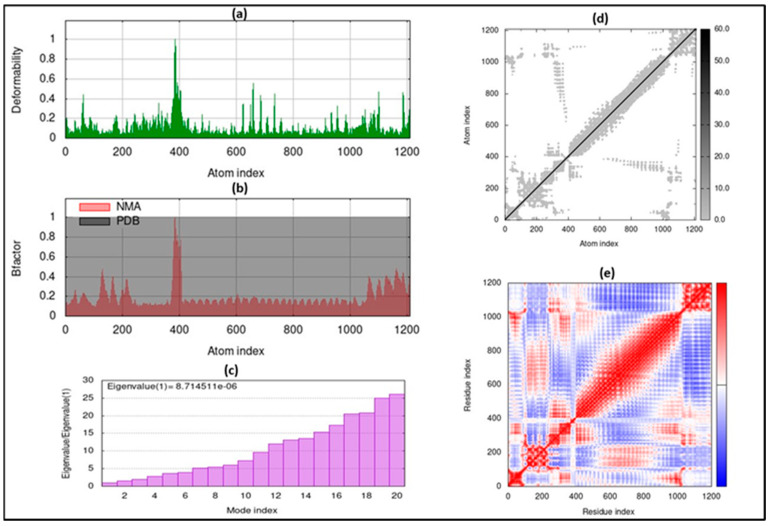
Molecular dynamic simulation of docked complex of vaccine and TLR5 showing (**a**) deformability; (**b**) B-factor; (**c**) eigenvalue; (**d**) elastic network analysis; (**e**) covariance matrix.

**Figure 5 vaccines-12-00139-f005:**
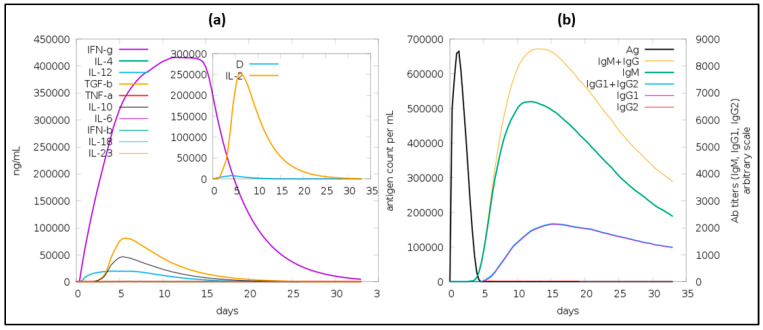
In silico evaluation of immune responses after vaccine injection. (**a**) Immunoglobulin generation; (**b**) production of B-cells after vaccine exposure.

**Figure 6 vaccines-12-00139-f006:**
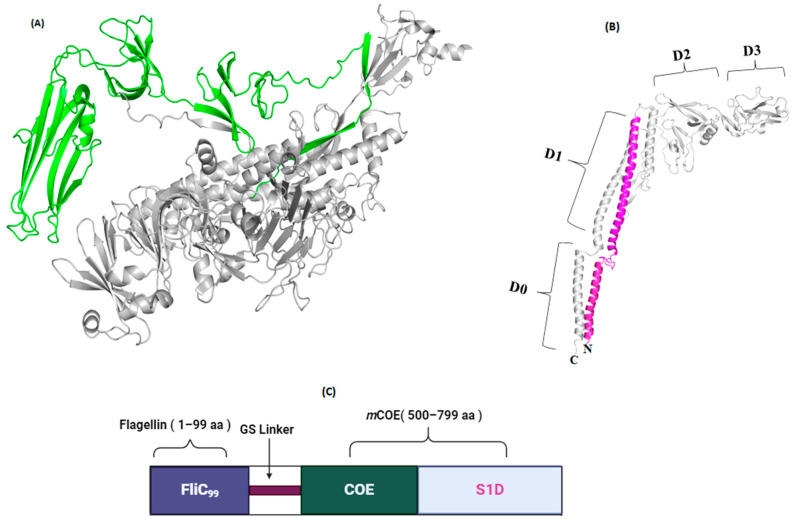
(**A**) Crystal structure of S-protein of PEDV, *m*COE, colored green, PDB ID: 7W6M. (**B**) The 3D structure of full-length flagellin (Protein Data Bank ID: 1UCU), FliC_99_, colored pink, as illustrated by the PyMol. (**C**) Construct of FliC_99_-*m*COE.

**Figure 7 vaccines-12-00139-f007:**
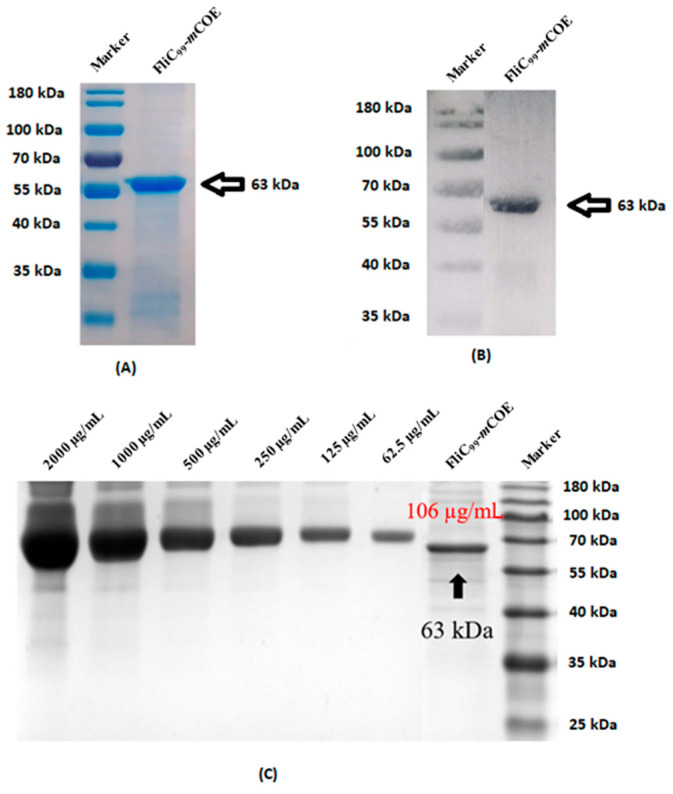
(**A**) SDS-PAGE of FliC_99_-*m*COE. (**B**) Western blot analysis of FliC_99_-*m*COE. (**C**) SDS-PAGE of FliC_99_-*m*COE quantification by different concentrations of bovine serum albumin (BSA) as standard.

**Figure 8 vaccines-12-00139-f008:**
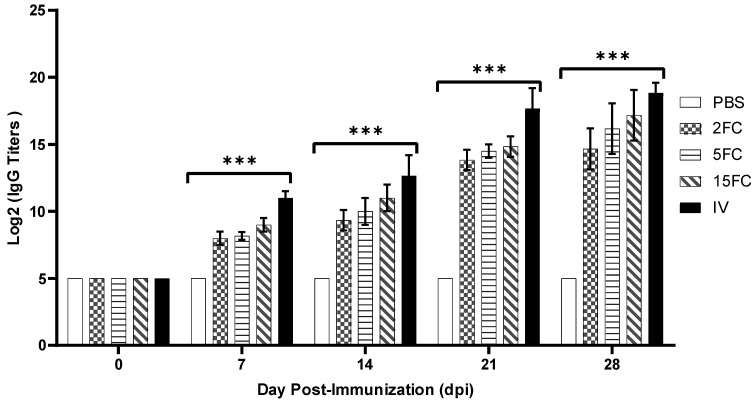
ICR female mice (n = 5 per group) were immunized intramuscularly at day 0 and 14 with five different formulations of vaccine. Groups vaccinated with FliC_99_-*m*COE: 2FC at 2 μg/mL, 5FC at 5 μg/mL, and 15FC at 15 μg/mL of the vaccine’s formulations. Inactivated vaccine (IV) group received 10^7^ TCID50/mL. Serum samples were collected at five different time points (0, 7, 14, 21, and 28 days) and antibody titers were analyzed by indirect ELISA coated with *inactivated PEDV*. All data are presented as mean ± SEM; *** indicates the significant differences (*p* < 0.05) between treatment groups at the same time point.

**Figure 9 vaccines-12-00139-f009:**
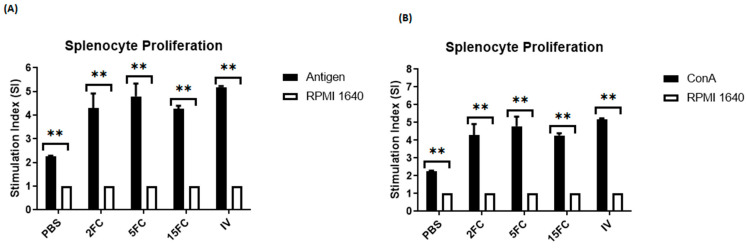
Splenocytes were taken from the immunized mice (n = 5 per group) of all groups. Groups vaccinated with FliC_99_-*m*COE: 2FC at 2 μg/mL, 5FC at 5 μg/mL, and 15FC at 15 μg/mL of the vaccine’s formulations. Inactivated vaccine (IV) group received 10^7^ TCID50/mL. Splenocytes were stimulated with *m*COE, ConA, and RPMI as negative control, and lymphocyte proliferation was evaluated by a CCK-8 kit. All data are presented as mean ± SEM and ** indicates significant differences (*p* < 0.05). (**A**) Lymphocyte proliferation activity of immunized mice splenocytes stimulated with antigen (*m*COE) and negative control. (**B**) Lymphocyte proliferation of immunized mice splenocytes stimulated with ConA and negative control.

**Figure 10 vaccines-12-00139-f010:**
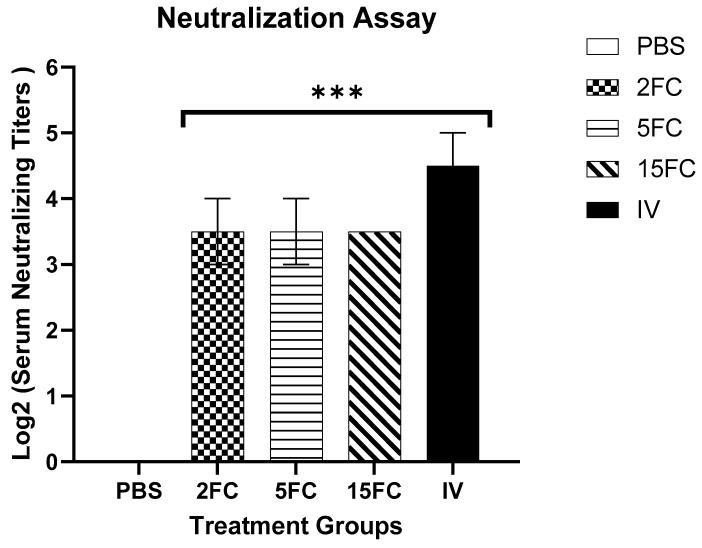
ICR serum samples were collected on day 28 after first immunization from mice (n = 5 per group), and 1000 TCID_50_ of PED virus was used to measure the neutralizing antibody levels. Titers of neutralizing antibodies were determined by the highest dilution that was able to neutralize PEDV on Vero cell lines. Groups vaccinated with FliC_99_-*m*COE: 2FC at 2 μg/mL, 5FC at 5 μg/mL, and 15FC at 15 μg/mL of the vaccine’s formulations. Inactivated vaccine (IV) group received 10^7^ TCID50/mL. All data are presented as mean ± SEM and *** indicates significant differences (*p* < 0.05) between treatment groups.

**Table 1 vaccines-12-00139-t001:** Designed primers for gene cloning purposes and for the construction of recombinant protein.

Target	Sequence (5′-3′)	RE	Gene-	DNA
Gene		Site	Length (bp)	Templates
FliC_99_	F ^1^ gg*gaattc*atggcacaagtcattaatacaaac	EcoRI	312	FliC
	R ^2^ **gctgccgcccccgcc**agactgaaccgccagttc	-		
*m*COE	F ^2^ **ggcgggggcggcag**ctccttcgtcaccttgcc	-	915	Modified S1S2
	R aa*ctcgag*cagctgcaagtactccgt	Xhol		
FliC_99_-*m*COE	F *ggatcc*aacccgctgcagaaaattg	EcoRI	1212	FliC_99_ and *m*COE
	R *ctcgag*cgtaacgacagaccctgt	Xhol		
T7 Promoter	F taatacgactcactataggg	-	705	
	R gctagttattgctcagcgg	-		

^1^ Italics in the primers represent restriction enzyme (RE) sites. ^2^ Bold fonts in the primers represent glycine–serine linkers.

**Table 2 vaccines-12-00139-t002:** Different parameters checked for the top TLR5–vaccine docked cluster, where RMSD stands for Root Mean Square Deviation.

Parameter	Value
HADDOCK score	480.6 ± 22.4
Cluster size	5
RMSD from overall lowest energy structure	12.9 ± 0.4
Van der Waals energy	−137.6 ± 4.3
Electrostatic energy	−255.8 ± 55.4
Desolvation energy	−3.8 ± 15.1
Restraints violation energy	6655.6 ± 182.22
Buried surface area	4504.2 ± 41.1
Z-score	−1.2

**Table 3 vaccines-12-00139-t003:** List of hydrogen bonds formed between vaccine and TLR5.

Hydrogen Bond Number	Vaccine		TLR5		Distance (Å)
	Amino Acid Residue	Residue Number	Amino Acid Residue	Residue Number	
1	ASP	43	TYR	608	2.92
2	ASP	43	SER	613	2.65
3	ASP	43	TYR	611	2.71
4	ARG	93	GLU	632	2.68
5	ARG	93	GLU	632	2.70
6	GLU	94	THR	622	3.01
7	ALA	100	GLN	595	3.31
8	CYS	254	ARG	604	2.75
9	ASN	272	PHE	25	3.20
10	ASN	272	ASP	26	2.77
11	SER	306	ASP	605	2.70
12	THR	329	TYR	534	2.95
13	THR	329	ASP	606	2.92
14	LEU	336	THR	10	2.87
15	PHE	339	THR	10	2.86
16	TYR	369	GLY	2	3.03

## Data Availability

The original contributions presented in the study are included in the article/Appendix A, further inquiries can be directed to the corresponding author.

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
