# Peer review of "Advancing PEDV Vaccination: Comparison between Inactivated and Flagellin N-Terminus-Adjuvanted Subunit Vaccines"

_vaccines, 2024, doi:10.3390/vaccines12020139_

Round 1

Reviewer 1 Report

Comments and Suggestions for Authors

This manuscript from Murtaza et al details the production and characterization of a novel recombinant subunit vaccine against Porcine Epidemic Diarrhea virus (PEDV).  PEDV remains a serious concern worldwide for the swine industry due to its high mortality rate and concerns over pathogenic infection and mutation from the use of current live-attenuated vaccines.  In the interest of expanding the repertoire of available PEDV vaccines, the authors engineered a recombinant protein consisting of a subunit of the PEDV spike protein (mCOE) linked to a subunit of flagellin (Flic99) previously found to act as an adjuvant by engaging with TLR5.  The recombinant vaccine (FliC99-mCOE) was predicted to be stable, immunogenic and to interact with TLR5 by computational modeling.  Vaccination of mice with FliC99-mCOE elicited humoral responses and lymphocyte proliferation comparable to IV inactivated PEDV (though slightly lower) that increased with a booster dose.  

As a whole, the article is well written and provides solid preliminary evidence for further testing of FliC99-mCOE in pigs.  My biggest comment is that the article should reference, either with original or published data, the immune responses resulting from mCOE or PEDV alone as an indication of the immunity boost elicited by conjugating to FliC99.  I also have the following minor comments to be addressed prior to acceptance:

1) In the Figure 3 description, the vaccine is represented in green instead of orange.  The authors should also specify that negatively charged amino acids are shown in red, and should clarify the difference between neutral and aliphatic amino acids (polarity).

2) The authors should consider including 1-2 more graphs in figure 5 to account for the smaller concentrations of some of the cytokines listed.

3) In figure 8, the authors should clarify whether the IV inactivated PEDV was significantly higher than the FliC99-mCOE groups at any time point.

Comments on the Quality of English Language

Minor editing of English language required in a few spots in the article.

Reviewer 2 Report

Comments and Suggestions for Authors

The Porcine Epidemic Diarrhea Virus (PEDV), which causes highly contagious diarrhea in suckling piglets with mortality rates reaching 100%, poses significant economic challenges. Vaccination is still the most effective method of preventing PEDV infection. Subunit vaccine development focuses on neutralizing antibody epitopes can overcome the disadvantages of traditional vaccines, particularly virulence reversion.

The authors of this manuscript created the FliC99-mCOE fusion protein, which consists of the modified COE region of the spike (S) protein and N-terminus of flagellin (FliC99). While similar concept studies have been conducted previously, the key novelty of this study is the comparison of immune responses between inactivated PEDV vaccine and flagellin-adjuvanted subunit vaccines. The result showed that FliC99-mCOE could induce immune responses against PEDV in mice that were comparable, though not superior, to those elicited by inactivated vaccine.

The authors conclude that “this study introduces a promising PEDV subunit vaccine combining a modified epitope (mCOE) with truncated flagellin, eliciting strong immune responses and showing high stability and safety” however, there is a lack of sufficient data to support the claims about the subunit vaccine’s stability and safety. Additional experiments, particularly challenge studies, are needed to support the author’s assertion that “In vivo studies in mice reveal efficacy comparable to inactivated vaccines.”

It would be interesting to see if the combination of multiple regions of neutralization epitopes could enhance the immunogenicity of the subunit vaccine. Furthermore, because mice are not the natural host or suitable for a PEDV challenge study, further research is needed to assess the immunoprotection of this subunit vaccine candidate. Additionally, there are some concerns need be addressed.

1. The title of this manuscript is not precise enough; it can be improved.

2. On page 2, line 1, the description "the S1D region of S2 as another target for neutralizing antibodies" is incorrect; S1D is in the S1 domain.

3. What is the final dose of each vaccine? The authors did not specify how many vaccines were injected into each mouse.

Reviewer 3 Report

Comments and Suggestions for Authors

This manuscript describes the investigation of a promising vaccine design strategy targeted at protecting swine from the alphacoronavirus PEDV. Although this virus is host-species specific, cannot infect humans or other animals, and presents no danger in contaminating food supplies, the economic losses to the swine industry are substantial given the high viral transmissibility and mortality rates. Vaccines in widespread use are inadequate to protect against emerging strains of PEDV, so new vaccine antigens with broadly conserved neutralizing epitopes should be investigated. The authors present here an early-stage assessment of a new subunit fusion vaccine candidate intended to boost the immunogenicity of conserved PEDv antigen epitopes.

Some of the most commonly used, premier in silico tools were not employed (e.g., AlphaFold 2 and DiscoTope 3.0, for tertiary structure and B cell epitope prediction, respectively). However, these are not essential given the depth of consilience and crossvalidation between results from an impressive battery of other excellent tools that were used. The in vivo assessments of cellular and humoral immunity constitute reasonable proxies for antigen immunogenicity, in line with expectations for initial examination of vaccine candidates.

In discussion, the authors appropriately point out that these results highlight the potential for their approach, and that further testing will be needed in the future, especially testing in animal challenge experiments, though that followup is beyond the scope of this early research.
